**Brief Investigation**

# Dynamic changes in neuronal and glial GAL4 driver expression during *Drosophila* aging

Caroline Delandre (ID) ,[1],*,† John P.D. McMullen (ID) ,[1],† Owen J. Marshall (ID) [1],*

[1]Menzies Institute for Medical Research, University of Tasmania, 17 Liverpool St, Hobart 7000, Australia

*Corresponding author: Menzies Institute for Medical Research, University of Tasmania, 17 Liverpool St, Hobart 7000, Australia. Email: owen.marshall@utas.edu.au;
*Corresponding author: Menzies Institute for Medical Research, University of Tasmania, 17 Liverpool St, Hobart 7000, Australia. Email: caroline.delandre@utas.edu.au
†These authors contributed equally to this work.

Understanding how diverse cell types come together to form a functioning brain relies on the ability to specifically target these cells. This is often done using genetic tools such as the GAL4/UAS system in *Drosophila melanogaster*. Surprisingly, despite its extensive usage during studies of the aging brain, detailed spatiotemporal characterization of GAL4 driver lines in adult flies has been lacking. Here, we show that 3 commonly used neuronal drivers (*elav[C155]-GAL4*, *nSyb[R57C10]-GAL4*, and *ChAT-GAL4*) and the commonly used glial driver *repo-GAL4* all show rapid and pronounced decreases in activity over the first 1.5 weeks of adult life, with activity becoming undetectable in some regions after 30 days (at 18°C). In addition to an overall decrease in GAL4 activity over time, we found notable differences in spatial patterns, mostly occurring soon after eclosion. Although all lines showed these changes, the *nSyb-GAL4* line exhibited the most consistent and stable expression patterns over aging. Our findings suggest that gene transcription of key loci decreases in the aged brain, a finding broadly similar to previous work in mammalian brains. Our results also raise questions over past work on long-term expression of disease models in the brain and stress the need to find better genetic tools for ageing studies.

Keywords: brain ageing; *Drosophila melanogaster*; GAL4/UAS system; gene expression; neuronal drivers; glial drivers; ageing research; FlyBase

## Introduction

Ever since the intricate drawings by Santiago Ramón y Cajal over a century ago (1904), the wide diversity of cell types populating the nervous system has presented a formidable challenge to neuroscientists. The genetic tools of *Drosophila melanogaster* have played an integral role in understanding the development of brain cell types and how they respond to aging and disease. The prime example is the GAL4/UAS bipartite gene expression system, in which cell-type-specific expression of the yeast transcription factor GAL4 can drive the expression of a transgene (Brand and Perrimon 1993). As most fly research has become heavily reliant on GAL4 lines, it is critically important to carefully describe their expression pattern, both spatial and temporal. Indeed, the chromatin environment of enhancers driving GAL4 expression can often be highly dynamic throughout development, and therefore, a specific GAL4 driver may target different cell types over time (Markstein *et al.* 2008). Anecdotal reports in the literature have indicated potential problems with widely used GAL4 lines, most prominently with the finding that *elav[C155]-GAL4*, long considered to be restricted to postmitotic neurons and a popular choice for pan-neuronal expression, was also expressed in neuroblasts and glia during early development, in addition to tissues outside the nervous system (Berger *et al.* 2007; Casas-Tintó *et al.* 2017; Weaver *et al.* 2020; Winant *et al.* 2024).

There is a knowledge gap in our understanding of the activity behavior of commonly used nervous system GAL4 lines in older flies, even though *Drosophila* is a powerful model organism to study the fundamental mechanisms of aging and neurodegenerative diseases (for review, see McGurk *et al.* 2015). Early work characterizing enhancer traps using beta-galactosidase already showed changes in expression patterns across lifespan (Helfand *et al.* 1995). These were supported by several reports from the Benzer and Seroude groups using a GAL4 enhancer trap set (Seroude 2002; Seroude *et al.* 2002; Poirier *et al.* 2008).

Another caveat is that a common readout for GAL4 driver expression is a stable fluorescent reporter driven through development, and the potential for its accumulation to mask the adult expression dynamic of the GAL4 driver. Moreover, immunostaining against induced reporter proteins, which can reduce the signal-to-noise ratio of fluorescent reporters (Wissing *et al.* 2022), and the chromatin environment of the reporter insertion site could also affect the final readout (Pfeiffer *et al.* 2010).

Here, we used the TARGET system (McGuire *et al.* 2003) to temporally restrict GAL4 driver activity and profile the expression of commonly used neuronal and glial driver lines during adult brain aging: *elav[C155]-GAL4* and *nSyb-GAL4* (also known as *R57C10-GAL4*), drivers widely used as pan-neuronal, *ChAT-GAL4*, a cholinergic neuronal driver, and the pan-glial driver *repo-GAL4*. We found that the expression patterns of both pan-neuronal drivers in young adult brains were not as uniform as expected, instead showing enrichment in particular brain regions. More importantly, we found there was a dramatic decrease in expression of all

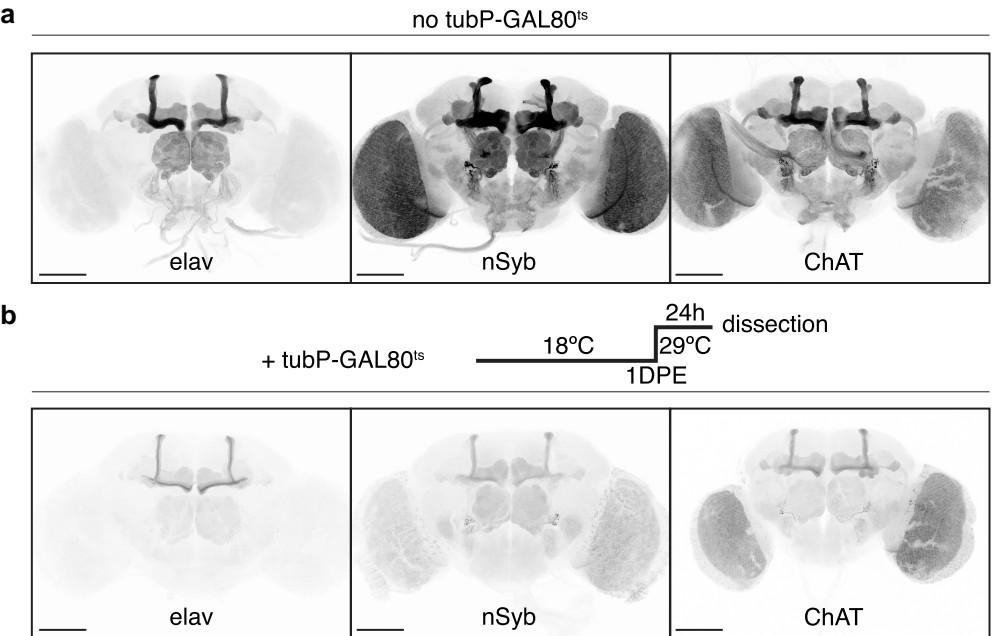

**Fig. 1.** Common neuronal GAL4 drivers are expressed differentially across the brain. a) 1-DPE brains were imaged for membrane-bound GFP driven by *elav[C155]-GAL4*, *nSyb[R57C10]-GAL4*, or *ChAT-GAL4*, either throughout development (A) or for 24 h right before dissection via a temperature-sensitive GAL80 (b). Scale bar, 100 μm.

drivers during the first days after eclosion, with expression continuing to decrease throughout aging.

## Materials and methods
### *Drosophila* maintenance and stocks

Flies were cultured in a standard food medium consisting of corn meal (35 g/L), dextrose (55 g/L), and yeast (50 g/L), and kept at 18°C or 25°C in incubators with controlled humidity (70%) and light: dark cycle of 12 h each. Drivers used were *P{GawB}elav[C155]* (*elav[C155]-GAL4*; BDSC_458), *P{GMR57C10-GAL4}attP2* (*nSyb-GAL4*; BDSC_39171), *Mi{Trojan-GAL4.0}ChAT[MI04508]* (*ChAT-GAL4*; BDSC_60317), and *P{GAL4}repo* (*repo-GAL4*; BDSC_7415). Green fluorescent protein (GFP) stocks were *10XUAS-IVS-myr::GFP* in attP2 (BDSC_32197), attP40 (BDSC_32198), or su(Hw)attP5 (BDSC_32199). Temperature-sensitive GAL4 inhibition was achieved using *P{tubP-GAL80[ts]}* either on chromosome 2 or 3 and keeping flies at 18°C. Flies were transferred to a 29°C incubator for 24 h to induce the expression of UAS reporters.

### Brain dissection and imaging

Adult brains were dissected in phosphate-buffered saline (PBS) and fixed in 4% paraformaldehyde (Electron Microscopy Sciences, USA) for 20 min, followed by 3 washes in PBS + 0.3% Triton X-100 and mounting onto slides with VECTASHIELD PLUS with DAPI (Vector Laboratories, USA). Slides were imaged with an FV3000 confocal laser scanning microscope (Olympus, Japan) under a 20X objective. Optimal laser settings were independently adjusted for each GAL4 driver to ensure proper spatial resolution. However, imaging settings were not changed when comparing time points. Four to eight brains were dissected for each time point and driver [except for *ChAT-GAL4* at 15 days posteclosion (DPE) with only 3 brains]. Since *elav[C155]-GAL4* is located on the X chromosome, only brains from flies of a specific sex were dissected (in this case, male flies).

## Analysis

Images were analyzed using the Icy open-source imaging software (de Chaumont *et al.* 2012) by creating regions of interest in selected brain areas from maximum projection images and recording the GFP mean pixel intensity. Values were plotted and logarithmic regression models were fitted using R (R Core Team 2021).

## Results
### Commonly used neuronal GAL4 drivers show differential spatial expression patterns in young adult brains

To test the reliability of brain GAL4 drivers throughout adult life, we profiled the expression of 3 commonly used neuronal drivers in young adult brains. The classic pan-neuronal *elav[C155]-GAL4* (Lin and Goodman 1994) is an enhancer trap line where GAL4 was inserted in the *elav* promoter region (Ogienko *et al.* 2020). A more recent pan-neuronal driver, *nSyb-GAL4,* was generated by cloning an enhancer fragment from the *nSyb* gene upstream of GAL4 and inserting into attP2 (chromosome 3) (Pfeiffer *et al.* 2008; Henry *et al.* 2012). The cholinergic driver *ChAT-GAL4* was engineered by inserting a GAL4 Trojan cassette into a MiMIC integration site located in an intron of the *ChAT* gene. This cassette places a T2A-GAL4 fragment flanked by splice acceptors so the GAL4 expression should reproduce the native pattern of *ChAT* (Diao *et al.* 2015). We obtained a baseline of their spatial expression profile by crossing these drivers to *UAS-myrGFP* (inserted in attP2) and culturing the progeny at 25°C until 1 DPE when they were dissected. The GFP expression pattern of both *elav[C155]-GAL4* and *nSyb-GAL4* was not as uniform across the brain as would be expected from pan-neuronal drivers; instead, we observed clear enrichment in specific areas, most notably the mushroom body (Fig. 1a). *elav[C155]-GAL4* showed the most dramatic differential expression, with very low signal in the optic lobes and midbrain areas other than the mushroom body (especially the alpha/beta

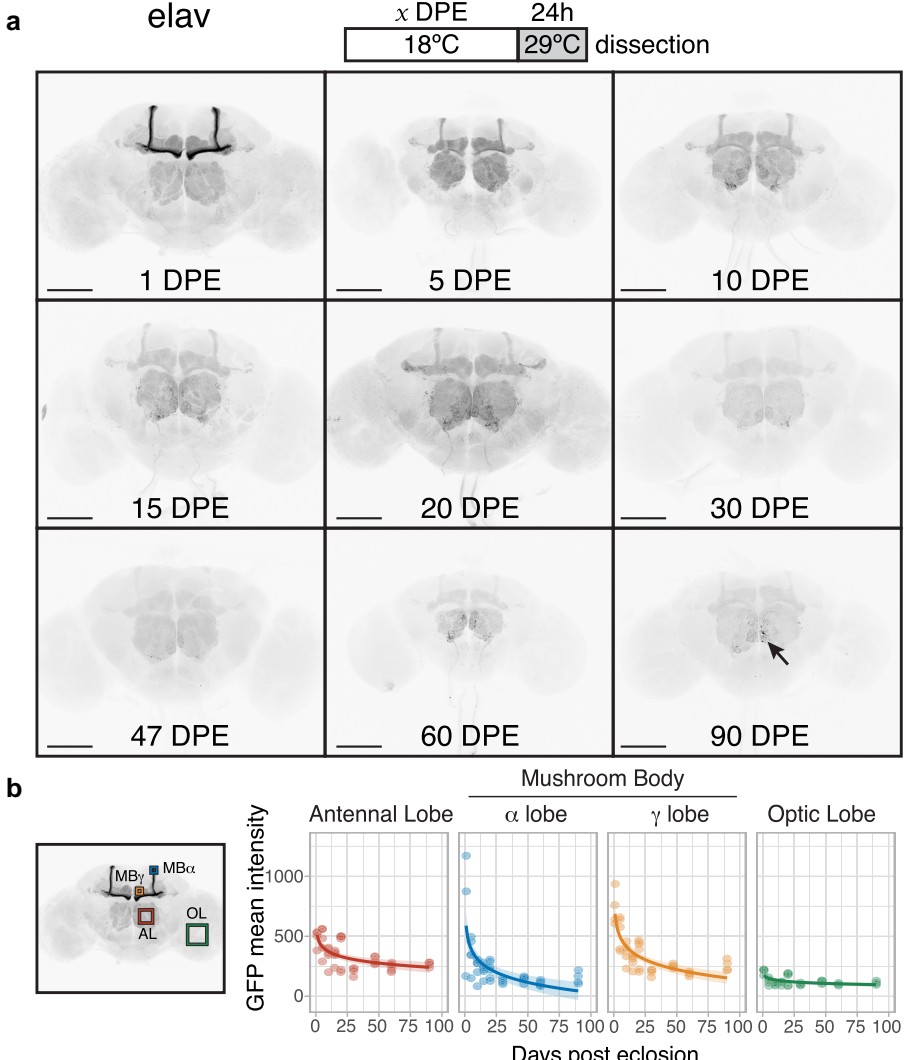

**Fig. 2.** a) Flies expressing membrane-bound GFP under the control of *elav[C155]-GAL4* and a temperature-sensitive GAL80 were induced for 24 h time windows by shifting from 18°C to 29°C at 9 time points throughout adult life (up to 90 DPE). Arrow indicates the appearance of individual neurons in older brains. b) Four brain regions were chosen for quantification of GFP mean intensity: the antennal lobe (AL), the α and γ lobes of the mushroom body (MBα and MBγ), and the optic lobe (OL) ($n = 4$–8). All regions showed a significant decline in driver activity; see Supplementary Table 1 for summary statistics of logarithmic regression models. Scale bar, 100 μm.

lobes) and the antennal lobes. *ChAT*-GAL4 also showed a mushroom body bias, but otherwise seemed to replicate the expression of cholinergic neurons (Salvaterra and Kitamoto 2001).

We then used the TARGET system to investigate how temporally restricting driver activity until eclosion would affect driver expression profiles. The TARGET system incorporates a temperature-sensitive GAL80 repressor under the *alphaTub84B* promoter (*tubP-GAL80[ts]*). In this system, GAL4 activity is inhibited at the restrictive temperature of 18°C, with full activity at the permissive temperature of 29°C (McGuire *et al.* 2003). Importantly, GAL4 activity is quickly restored in this system at the permissive temperature, with full GAL4/UAS activation of reporter transgenes observed within 6 h of the temperature switch (McGuire *et al.* 2003), making this method ideal for capturing temporal driver expression profiles. Indeed, the brains that were induced at 1 DPE showed a decrease in overall GFP signal (Fig. 1b). Spatial expression profiles were overall similar to images from brains that were not temporally restricted, indicating these GAL4 lines are likely driving expression in the same subset of neurons during the few days before eclosion. Taken together, these

results show that both the *elav*-GAL4 and *nSyb*-GAL4 pan-neuronal drivers do not drive expression ubiquitously across neuronal cell types, and that temporally restricting their activity provides a more accurate representation of GAL4 activity levels at that time point.

## Neuronal driver activity decreases throughout adult life in a brain region-specific manner

We then asked whether the expression profiles of neuronal drivers change as brains age. Again using the TARGET system, we restricted GAL4 activity to nine 24 h time windows by rearing flies at 18°C for 1–90 DPE and then shifting to 29°C for 24 h before brain dissection (Fig. 2). As life span is approximately doubled at 18°C compared to 25°C (Huang *et al.* 2020), 90 DPE is the equivalent of approximately 45 DPE at 25°C. All 3 neuronal drivers exhibited significant and substantial declines in activity (Figs. 2–4; Supplementary Table 1). For *elav[C155]-GAL4*, we noticed a reduction in driver activity starting at just 5 DPE (at 18°C) in the mushroom body alpha/beta lobes (Fig. 2). The mushroom body alpha′/beta′ lobes together with the optic lobes became barely detectable from 5 to 10 DPE.

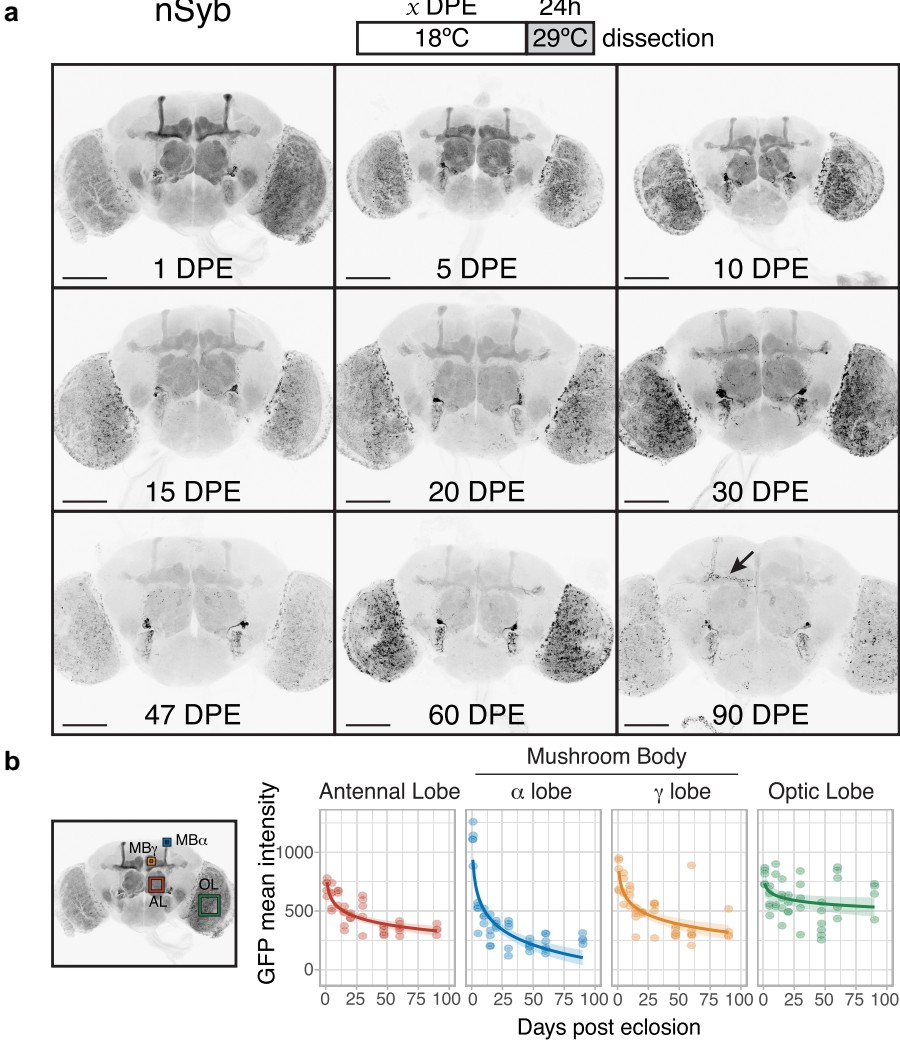

**Fig. 3.** a) Flies expressing membrane-bound GFP under the control of *nSyb[R57C10]-GAL4* and a temperature-sensitive GAL80 were induced for 24 h time windows by shifting from 18°C to 29°C at 9 time points throughout adult life (up to 90 DPE). Arrow indicates the appearance of individual neurons in older brains. b) Four brain regions were chosen for quantification of GFP mean intensity: the antennal lobe (AL), the α and γ lobes of the mushroom body (MBα and MBγ), and the optic lobe (OL) (*n* = 4–8). All regions showed a significant decline in driver activity; see Supplementary Table 1 for summary statistics of logarithmic regression models. Scale bar, 100 μm.

Moreover, there was a gradual decrease in overall driver activity in all profiled regions, notably during the first 30 days.

Similarly, we observed a gradual decline, albeit not as steep, in *nSyb-GAL4* activity during the first 30 days of adulthood (Fig. 3; Supplementary Table 1). There was also a reduction in the signal difference between the alpha/beta and gamma lobes: from 30 DPE onwards, the beta lobes were almost indistinguishable from the gamma lobes. The expression in the optic lobes gradually lost its uniform pattern as brains aged, with GFP aggregates most obvious in the oldest brains. Interestingly, a small group of neurons located next to the antennal lobes remained strongly labeled throughout the experiment. *ChAT-GAL4*, but not *elav[C155]-GAL4*, also drove persistent expression in this neuronal subset (Fig. 4a), ruling out a potential artifact specific to the *nSyb-GAL4* construct design.

*ChAT-GAL4* offered the most striking change in expression pattern over the initial DPE, most notably in the optic lobe, where GFP intensity decreased to almost a third of the 1-DPE baseline (Fig. 4). The mushroom body transitioned from a strong predominance of alpha/beta lobes (1 DPE) to an equal level of both alpha/beta and

gamma lobes (5 DPE), followed by a stage where only the gamma lobes had detectable signal (30 DPE). The remaining cholinergic regions followed a gradual decline in driver expression; the optic lobes remained the main visible areas at 90 DPE, in addition to the subset of neurons seen with *nSyb-GAL4* and another group of neurons projecting to the mushroom body. We also noticed the seemingly random appearance of individual neurons in older brains with all 3 drivers (for example, see arrows in Figs. 2a and 3a). We observed a similar expression pattern in 50-DPE brains from uninduced flies, implying the GAL4 activity present in older brains is likely due to a lack of GAL80 repression (Supplementary Fig. 1). This was especially noticeable in the optic lobes for *nSyb-GAL4*.

## Reporter levels decline with age regardless of the targeted insertion site

Previous studies have suggested that differences in expression levels or distribution could be affected by the insertion site of the reporter transgene, raising the possibility that the local chromatin environment of the reporter, rather than the activity of the driver,

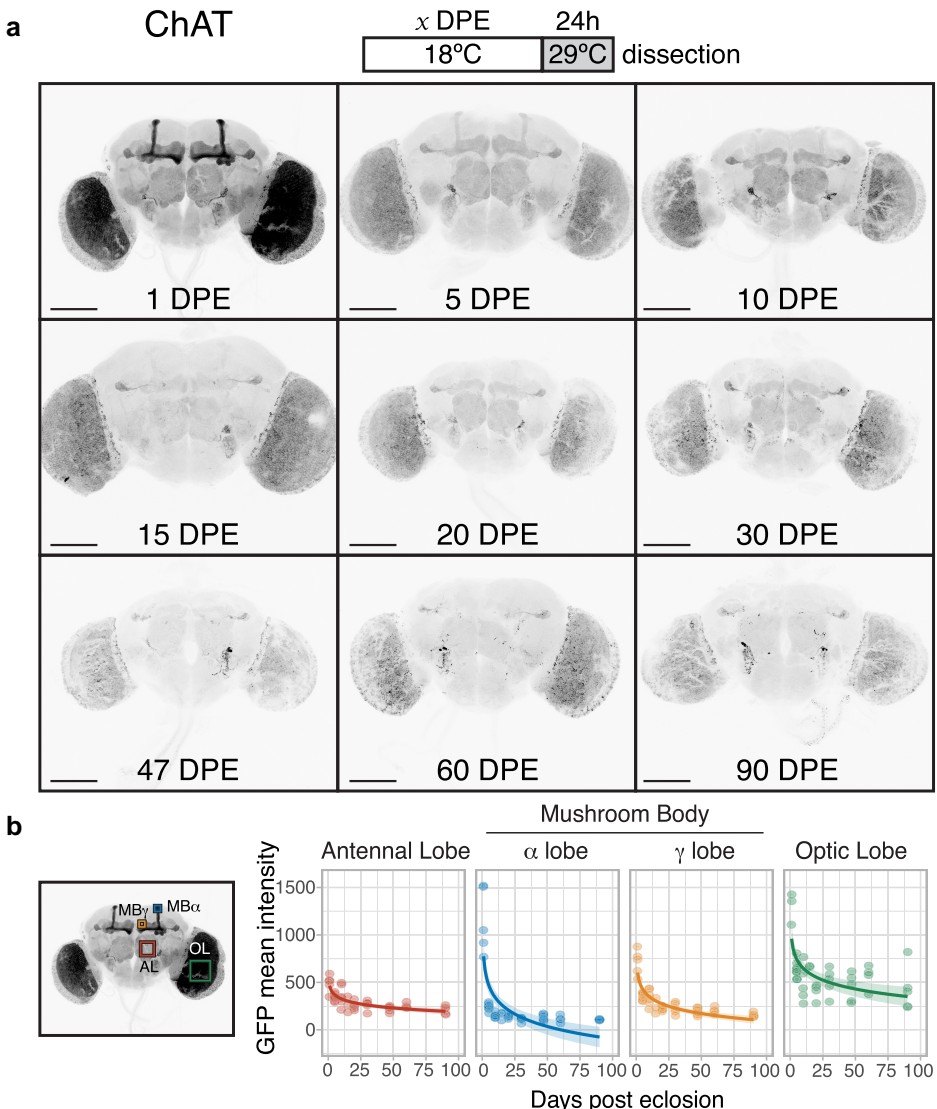

**Fig. 4.** a) Flies expressing membrane-bound GFP under the control of *ChAT*-GAL4 and a temperature-sensitive GAL80 were induced for 24 h time windows by shifting from 18°C to 29°C at 9 time points throughout adult life (up to 90 DPE). b) Four brain regions were chosen for quantification of GFP mean intensity: the antennal lobe (AL), the α and γ lobes of the mushroom body (MBα and MBγ), and the optic lobe (OL) ($n = 4$–8; except 15 DPE, with only 3 brains). All regions showed a significant decline in driver activity; see Supplementary Table 1 for summary statistics of logarithmic regression models. Scale bar, 100 μm.

might be affecting an age-related decline in activity (Markstein *et al*. 2008). To test this, we repeated the same time course with identical *UAS-myrGFP* transgenes inserted into two other targeted insertion site loci: attP40 (chromosome 2L) or su(Hw)attP5 (chromosome 2R). As previously published by the Rubin lab, we found baseline differences in levels and distribution of the membrane-bound GFP reporter among insertion sites at 1 DPE (Pfeiffer *et al*. 2010), with the most variability observed in the neurons of the optic lobe. However, the insertion sites did not affect the steep rate of decline over the first 25 DPE (at 18°C). Our data indicate that the age-related decline in neuronal reporter activity stems solely from the GAL4 driver line (Fig. 5, Supplementary Figs. 2–7 and Table 1).

### Glial driver activity similarly declines during brain ageing

Transcriptomic studies have found a decline in the expression of neuronal functional genes with aging, but also a concomitant

increase in the glial expression of specific genes (for review, see Ham and Lee 2020). However, a recent study found the activity of 2 glial subtype GAL4 drivers remained unaltered in older flies (Sheng *et al*. 2023). To determine whether the decline in gene expression is specific to neurons, we profiled the activity of the pan-glial driver *repo*-GAL4, derived from a P-element insertion into the *repo* promoter region, at 1 and 10 DPE (at 18°C). We observed a similar, significant decrease in expression at 10 DPE (Fig. 6).

Taken together, our data indicate that the expression profiles of multiple keystone cell-type defining GAL4 drivers change throughout adult life, both in intensity and region specificity.

## Discussion

GAL4 drivers have provided fly neuroscientists with powerful means to label and control brain cell types, with pan-neuronal lines such as *elav[C155]-GAL4* and *nSyb-GAL4*, pan-glial lines such as *repo-GAL4*, or more specific subtypes, such as cholinergic

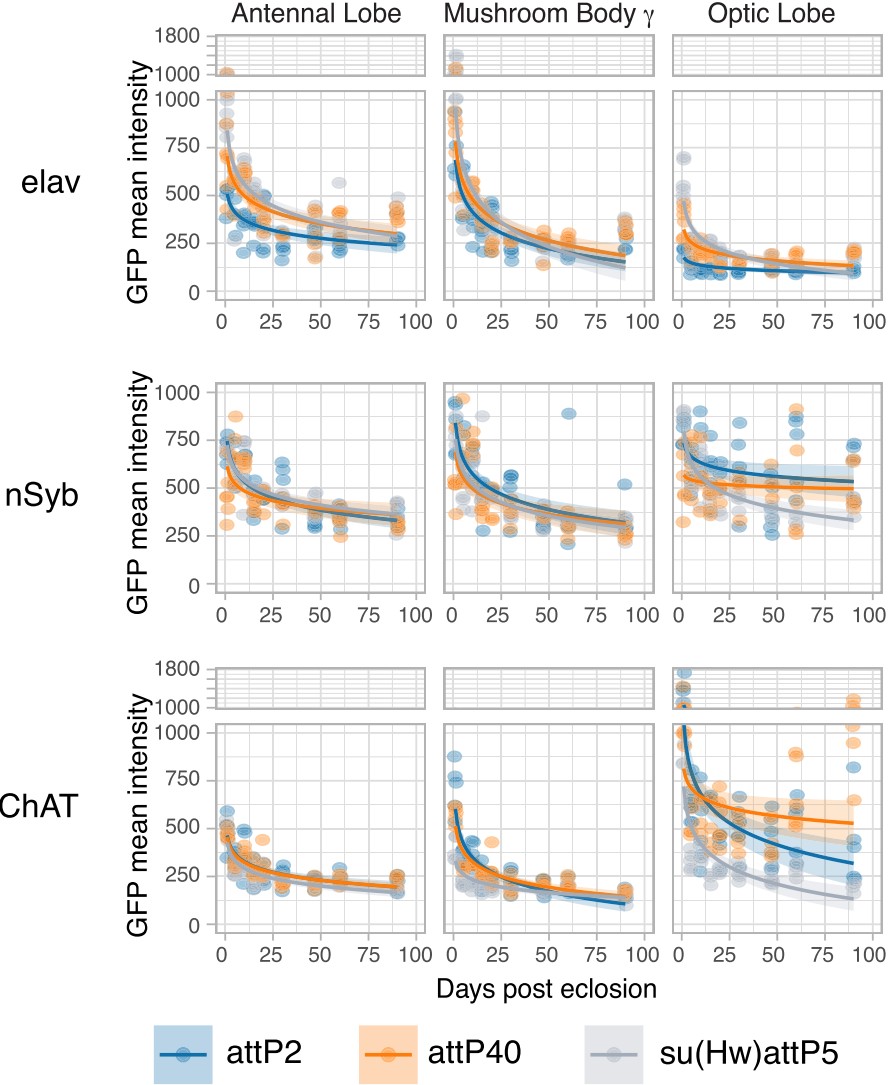

**Fig. 5.** Flies expressing membrane-bound GFP (inserted in attP2, attP40, or su(Hw)attP5) under the control of *elav[C155]-GAL4*, *nSyb[R57C10]-GAL4*, or *ChAT-GAL4* and a temperature-sensitive GAL80 were induced for 24 h time windows by shifting from 18°C to 29°C at 9 time points throughout adult life (up to 90 DPE). Quantification of GFP mean intensity in 3 brain regions is shown (representative brain images are in Supplementary Figs. 2–7). All regions showed a significant decline in driver activity, with the exception of *nSyb-GAL4* driving attP40 in the OL; summary statistics of logarithmic regression models in Supplementary Table 1.

neurons with *ChAT-GAL4*. Even though these drivers have become standards in the field and have been used extensively in adult fly studies, their expression patterns in the adult brain have been little investigated, especially during aging. We, therefore, decided to investigate their expression profile over multiple time points in adulthood. We found both pan-neuronal drivers to have distinctly different spatial patterns already in young adults. Moreover, while both showed a bias toward the mushroom body, only *nSyb-GAL4* showed strong expression in most other areas of the brain. As *elav[C155]-GAL4* is the classic pan-neuronal driver in the field (Lin and Goodman 1994), we were expecting a more uniform expression pattern across the brain. However, a close look at the literature showed similar images with low signal in the optic lobe compared to the mushroom body (especially the alpha/beta lobes) and antennal lobe (recent examples: Lin *et al.* 2015; Jonson *et al.* 2018; Chakravarti Dilley *et al.* 2020; Hawley *et al.* 2023).

In addition to expression pattern changes during early development, we found an age-dependent decline in expression activity in all drivers. Seroude and colleagues have shown over 80% of a set of

180 GAL4 enhancer trap lines change expression activity over time as flies age (Seroude 2002; Seroude *et al.* 2002). However, it was difficult to find similar studies performed on more widely used neuronal drivers. Only anecdotal findings, such as by the Thor lab, observe a decrease in driving activity of *elav[C155]-GAL4* in older flies (up to 20 DPE at 25°C); while levels of another pan-neuronal driver (also based on *nSyb*, but derived from a P-element insertion in an unknown locus) remain constant or even increase as flies age (Jonson *et al.* 2018, 2015). However, these studies did not restrict GAL4 activity to a narrow time window, making it difficult to interpret the expression dynamics of these drivers.

The first 5 DPE (roughly equivalent to 2.5 DPE at 25°C) showed the steepest decrease in intensity. Many experiments are performed within this time frame; however, most articles report the expression pattern of drivers in young flies without temporal restriction. By temporally restricting expression, we were able to reveal spatial pattern changes in the *ChAT-GAL4* and *elav[C155]-GAL4*-driven brains: most notably, a dramatic decrease in expression specifically in the mushroom body (for both drivers)

**Fig. 6.** Flies expressing membrane-bound GFP under the control of *repo*-GAL4 and a temperature-sensitive GAL80 were induced for 24 h time windows by shifting from 18°C to 29°C at 1 or 10 DPE. A central brain region was chosen for quantification of GFP mean intensity. Boxplot and individual values are shown ($n = 4$–6; $P = 0.0016$, unpaired Student's t-test). Scale bar, 100 μm.

and the optic lobe (for *ChAT*-GAL4). Overall, *nSyb*-GAL4 seemed the most consistent driver, with the signal still present in most brain regions until 30 DPE (18°C), after which it was significantly reduced. Although our findings illustrate the relative levels of transgene expression between regions, we note that 29°C induction for 24 h might not be long enough to observe the full spatial expression profile of a driver in older brains.

Every driver we have checked has shown a decline in expression over time, correlating with previous transcriptomic studies reporting a general age-dependent decrease of gene expression, especially in neurons (Lu *et al.* 2004; Erraji-Benchekroun *et al.* 2005; Berchtold *et al.* 2008). However, measuring the relative levels of transcripts could lead to misinterpretations because it relies on the assumption that overall RNA levels across samples are similar (Lovén *et al.* 2012). When measuring total RNA as flies age, older studies found a decline in RNA levels, especially steep right after eclosion (Tahoe *et al.* 2004), postulated to be due to the transition between metamorphosis, requiring high gene expression levels, and adulthood (Shikama and Brack 1996). More recently, a similar early decline was reported in the fly brain using single-cell transcriptomics (Davie *et al.* 2018; Lu *et al.* 2023). Our results provide an alternative approach to confirm the decrease of absolute gene expression at a region-specific level.

This global decline in transcription could also affect the activation of the UAS reporter by GAL4 itself. Our readout cannot differentiate between the transcription of *GAL4* vs *UAS-myrGFP*; however, the region-specific changes we observed differed between drivers, implying that these results are mainly caused by the enhancer-GAL4 element.

To the best of our knowledge, this is the first study investigating the expression profiles of commonly used brain GAL4 drivers over multiple temporally restricted time points in aging brains. Our findings highlight the variability of their spatiotemporal activity, even in young adult flies. *nSyb*-GAL4 had the most consistent expression profile across the brain in adult flies, making it a preferable GAL4 for long-term pan-neuronal expression in adults. However, all drivers were very weak after only 30 DPE at 18°C. This lack of reliable driver activity later in life implies phenotypes observed with current neurodegenerative disease models may be mainly due to the effect of a transgene during development or soon after eclosion, and results need to be interpreted accordingly. To study the interaction of aging and neurodegeneration, where the protein of interest is only produced in older brains, new driver systems will need to be developed. Our results also have implications to the broader fly community, as they underscore the critical importance of characterizing GAL4 driver lines in detail during experimental design, and we hope this study will encourage other researchers to add to this resource.

## Data availability

Fly stocks used in this study are available from the Bloomington Drosophila Stock Center (NIH P40OD018537) and imaging raw data from the corresponding authors upon request.

Supplemental material available at GENETICS online.

## Acknowledgments

We thank Jake Newland for helpful comments on the manuscript.

## Funding

This work was supported through NHMRC grants APP1128784 and APP1185220, Ian Potter Foundation grant 20190091, and philanthropic funding from the Menzies Institute for Medical Research, to O.J.M.

## Conflicts of interest

The author(s) declare no conflict of interest.

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

*Editor: K. Kaun*