## [Peer Review File · Genetics]

Dynamic changes in neuronal and glial GAL4 driver expression during *Drosophila* ageing

Caroline Delandre, John McMullen, and Owen Marshall

NOTE: The reviews and decision letters are unedited and appear as submitted by the reviewers.

In extremely rare instances and as determined by a Senior Editor or the EIC, portions of a review may be redacted. If a review is signed, the reviewer has agreed to no longer remain anonymous.

The review history appears in chronological order.

Review Timeline:

Submission Date:	2024-07-22
Editorial Decision:	2024-08-25
Appeal Received:	2024-09-20
Appeal Accepted:	2024-10-07
Revision Received:	2024-12-09
Accepted:	2024-12-19

MS ID#: GENETICS-2024-307300

Dear Dr. Marshall :

Your manuscript entitled "Dynamic changes in neuronal and glial GAL4 driver expression during *Drosophila* ageing" has now been reviewed. You can view the reviewers' comments below. Based on the reviewers' comments, I have concluded that your manuscript is not acceptable for publication in GENETICS.

However, because we feel that your manuscript presents data that are likely useful and of interest to the community, we have consulted with the G3 editorial board and recommend that you consider submitting a suitably revised manuscript for consideration for publication in our sister journal G3: Genes|Genomes|Genetics (<https://www.g3journal.org>). Please note that submission to G3 does not guarantee acceptance; G3 will provide you with an independent editorial decision.

The reviewers raise important critiques that you should consider addressing. For example, both reviewers are concerned that 24 hours at 29 C may not be sufficient for Gal4 to be turned on. Similarly, myself and the reviewers are surprised that you could get flies to live out to 90 days, especially at that temperature. There are a number of other reasonable criticisms that the G3 editorial board thinks are fair and need to be addressed.

G3 agrees to conditionally accept the manuscript, as long as you are able to respond to the full set of critiques. If you decide to transfer please ask for Dr. Michelle Arbeitman as Senior Editor. If the option to request me as Associate Editor is provided, please enter my name as well. G3 may decide to re-review, depending on the responses.

If you choose to transfer your manuscript, please revise as requested above and provide a point-by-point response to reviews when re-submitting. The original reviews and GENETICS decision letter will be made available to the G3 editors.

As an open-access journal published by the Genetics Society of America, G3 seeks to publish manuscripts reporting useful data and resources, regardless of the perceived significance or potential size of the audience. The quality of the science, and the standard to which scientists are held, is exactly the same between the two journals and both journals have a rigorous policy on the availability of data. G3 has a mandate to publish foundational research that is useful to geneticists, while GENETICS is focused on publishing work that will be of broad interest and that provides clear new insight relevant to our discipline.

Thank you again for your interest in GENETICS. We hope you will transfer your paper to G3 and will not be deterred from submitting future manuscripts to GENETICS.

To begin the transfer process to G3, click the link below:

Link Not Available

Sincerely,
Karla Kaun
Associate Editor
GENETICS

Approved by:
Kate O'Connor-Giles
Senior Editor
GENETICS

Reviewer #2 (Comments for the Authors (Required)):

See attachment

Reviewer 2 Attached Comments:

This is an important study addressing an issue that has been largely neglected by many labs that are using the UAS/GAL4 system to examine the function of a gene during aging.

The authors raise the additional issue that the measurements of expression levels with a fluorescent reporter can be affected by protein accumulation. The authors elegantly overcome this problem by repressing the activity of GAL4 with GAL80. The experimental results clearly show that the use of GAL80 reveal a more accurate description of the expression pattern of the GFP reporter by a GAL4 driver. This issue does not apply when using a lacZ reporter which has been shown by the Helfand lab to have ~8h half-life in vivo in aging animals (Mech Dev 55(1): 45-51, 1996). I wonder if the half-life of fluorescent reporters has been previously measured.

The experiments are carefully done and described and support the conclusions of the authors. The only thing that the authors have not considered and should be mentioned in the discussion is the fact that GAL80 inactivation may not be complete and therefore could lead to an under-estimation of expression. Although (to my knowledge) it has not been investigated with neuronal/brain drivers, 24h at 29°C does not result in full induction with muscle drivers (microPublication Biology 2023. 10.17912/micropub.biology.000770.).

The first section of the results would benefit of some clarifications. ONLY the C155 driver is known to be described as pan-neuronal while ChAT and nSyb are not as far as I know. It is unclear what drivers the authors refer to as pan-neuronal when they write: "The GFP expression pattern of both pan-neuronal drivers was quite different from what we expected".

The reference de Chaumont et al., 2012 is not in the reference list.

Only figure 1 shows the magnification scale, it is missing in Figures 2, 3, 4, 6 and all supplemental figures.

The Data Availability Statement is missing (should be at the end of Material & Methods).

Reviewer #3 (Comments for the Authors (Required)):

In the manuscript by Delandre et al, the authors describe changes in neuronal and glial GAL4 driver expression with age in *Drosophila*.

Given that careful characterization of tools is key for the correct interpretation of results, this work is clearly relevant for other researchers using *Drosophila* in studies of ageing-associated phenotypes and neurodegeneration.

I have the following comments (in order of appearance in the manuscript):

1. In the second last paragraph of the introduction, the authors state that *Drosophila* is becoming a powerful model organism ... I think it is better to rephrase this since *Drosophila* has been used for these types of studies since at least two decades.
2. In the last paragraph of the introduction, the authors write that the expression patterns in young adult brains were surprisingly not as uniform as expected. This statement is based on the assumption that it should be uniform, but is the evidence for that? *Elav* starts to be expressed in newly differentiating postmitotic neurons, and it may well have an initially higher expression that then goes down to a maintenance level. In other words, it may well be that this is the correct reflection of expression dynamics (that may not have been appreciated until now). At any rate, I suggest to carefully check the literature and possibly modify this statement.
3. The authors use a shift to 29°C to induce expression of UAS reporters and this for 24 hours in all conditions. It is not clear to me whether the authors have checked the induction dynamics. Is 24h sufficient to get full expression (we typically use 48-72hrs of induction)? Is it certain that the induction dynamics itself does not change with age (certainly in light of overall reduced transcription at higher age)?
4. Please include in the materials and methods how long the brains were fixed. Since the authors do not perform antibody staining to visualize GFP, this information is important. Prolonged fixation tends to reduce fluorescence emitted by GFP.
5. I do not understand why the fact that *elav*-GAL4 is located on the X chromosome is an argument to only dissect male brains. One can equally well generate females.
6. Did the authors use single optical sections in their analyses? Or did they use Z-stacks? Please clarify. If single optical sections were used, how was position in the brain controlled for?
7. In the first part of the results, the authors write: 'Flies were reared at 18°C until 1DPE when brains were placed at 29°C ...' Were the brains first dissected and then placed at elevated temperature?
8. It is remarkable that the authors were able to look at flies at age 90DPE. Usually, maximal life expectancy is about 60 days. Please explain.
9. Glial driver activity. The authors refer to the study by Sheng et al. Which glial GAL4 drivers did Sheng et al study? Did it include *repo*-GAL4? If not, how can *repo*-GAL4 be a good comparison?
10. The authors use the Wissing et al (2022) reference, but as I understand it, this is a mouse reference. Are there no equivalent references for *Drosophila*?
11. In the discussion, the authors refer to another pan-neuronal driver (also based on *nSyb*). Please include the reference!
12. It would help the manuscript, if the authors would analyze the single cell expression data that were recently published for expression of *elav*, *nSyb*, *ChAT*, *repo*, and *alphaTub84B* and determine how this changes with age.
13. In the final part of the discussion, the authors comment on all GAL4 drivers being very weak after only 30DPE at 18°C. Is this really so important? The majority of studies using *Drosophila* as model for neurodegeneration do not use flies older than 30 days, but more typically up to 21 days. Up until 28 days of age, flies tend to be fine, but then a fairly rapid decline in viability starts, making experimentation much more troublesome.

Associate Editor Comments: (please see above)

CRITERIA FOR PUBLICATION IN GENETICS:

GENETICS considers for publication manuscripts that are of general interest to a wide range of genetics and genomics investigators or of extraordinary interest to specialists. The results presented must provide strong support for the conclusions reached. The study must also provide significant new insights into a biological process, or demonstrate novel and creative approaches to an important biological problem, or describe development of new resources, methods, technologies, or tools of interest to a wide range of geneticists.

CRITERIA FOR PUBLICATION IN G3

- the experiments and other analyses are of high-quality and are clearly described in sufficient detail to reproduce the results, which are useful to the community;
- the study describes new data and information (e.g. genome assemblies, RNA-seq, association studies), reagents or new resources (e.g. results of a mutant screen; mutant; collections for functional genomic experiments) or novel tools/methodologies (e.g. statistical/computational methods) whose publication would be valuable for researchers and other stakeholders;
- the results are original and all community standards for data availability and format are followed;
- the results presented provide strong support for the conclusions reached.

Dr Owen Marshall
Menzies Institute for Medical Research
17 Liverpool St
Hobart, Tas, 7000
Australia

September 20, 2024

Dear Dr Kaun,

We were surprised by the decision to reject our manuscript from GENETICS, especially given the positive nature of the reviews and the initial favourable response from the senior editorial team. We believe there might have been a misunderstanding regarding our experimental setup, particularly concerning the temperature and ageing dynamics. We hope the following clarifications will prompt you to reconsider your decision.

First, we would like to clarify that all ageing experiments were conducted at 18°C, not 25°C as may have been inferred. At this temperature, *Drosophila* developmental timing and lifespan are doubled when compared to 25°C (Huang *et al*, *PLoS Biol*, 2020), making it typical for flies to survive to 90 days post-eclosion (DPE) and beyond. This is a critical point, as it aligns our observations of GAL4 driver dynamics with the lifespan of *Drosophila* used in neurodegenerative models, including those for Alzheimer's, Parkinson's, and ALS. We acknowledge that this was not sufficiently emphasised in our manuscript, and we have revised the text to clarify these dynamics. This clarification underscores the direct relevance of our findings to a broad body of literature and highlights the importance of disseminating our findings to inform future research protocols on these diseases.

Regarding GAL4 induction dynamics, we used consistent induction timings throughout our experiments. In the TARGET system, GAL4 is continuously expressed and bound at the UAS enhancer at 18°C, with GAL80^{ts} binding and masking the GAL4 activation domain to prevent gene transcription. Upon a temperature shift, the activation domain is unmasked, and UAS transgene transcriptional induction is immediate, accurately reflecting GAL4 levels (see the original TARGET publication, McGuire *et al*, *Science*, 2003, which demonstrates mRNA equivalence of a GFP reporter gene after 6 hours of induction). A longer induction time would only result in more accumulation of our fluorescent reporter, rather than reflecting any change in induction dynamics, while reducing the temporal resolution of our data. Our approach captures the immediate levels of GAL4 induction at different ageing time points, which is crucial for understanding GAL4 driver use during ageing.

A detailed response to all reviewer comments follows after this letter.

Given these clarifications, we respectfully request that GENETICS reconsider its decision. We believe our findings are of significant interest to the community and provide valuable insights into the use of genetic tools in ageing research, as well as new evidence for a general decrease in gene expression during ageing.

Thank you for considering our appeal.

Sincerely,

Owen Marshall

October 7, 2024

Dear Dr. Marshall,

Thank you for your patience. As a peer-edited, society journal, GENETICS takes all appeals very seriously. We have studied your appeal letter, response to reviews, the original reviews and manuscript. We are pleased to let you know that your appeal has been accepted and a revised version of your manuscript will be considered for publication in GENETICS as a Brief Investigation.

In your revisions please address the following (and be sure to highlight the changes to your original manuscript):

(1) Further explanation of the methodology used is needed to provide clarity to the data and help explain the longevity of your flies. This includes a more thorough description of the temperature switches you chose, as well as the length of induction.

(2) We agree with the reviewers that one of the reasons for the lack of broad expression you see is probably due to the short induction time used in your methodology. While your 24 hr induction period is consistent across all the timepoints tested and provides rigorous data that will be helpful for the community, it is also likely that a longer induction time could provide broader expression. It is important that you discuss this in your manuscript.

(3) Please ensure that your manuscript fits within the word count guidelines for a Brief Investigation. Brief Investigations should not exceed 3000 words, excluding abstract, tables, figure legends, and literature cited, with figures and tables limited to the fewest necessary to describe the results. We believe you could significantly reduce the text in your Introduction and condense the points in your Discussion.

Please use the following link to submit your revised manuscript files:

<https://genetics.msubmit.net/cgi-bin/main.plex?el=A7NR5GXe6A2Prk7I3B9ftdo3AX8aE6A8BFpZuneEyTkAZ>

Thank you for your interest in publishing in GENETICS!

Sincerely,

Howard Lipshitz
Editor in Chief
GENETICS

Kate O'Connor-Giles
Senior Editor
GENETICS

Karla Kaun
Associate Editor
GENETICS

Response to reviewers

We thank the reviewers for their helpful comments and suggestions. A detailed response to the points raised by each reviewer is provided below.

Reviewer #2

The experimental results clearly show that the use of GAL80 reveal a more accurate description of the expression pattern of the GFP reporter by a GAL4 driver. This issue does not apply when using a lacZ reporter which has been shown by the Helfand lab to have ~8h half-life in vivo in aging animals (Mech Dev 55(1): 45-51, 1996). I wonder if the half-life of fluorescent reporters has been previously measured.

We also wondered the same question, but could not find any study characterising the half-life of myristoylated GFP.

The only thing that the authors have not considered and should be mentioned in the discussion is the fact that GAL80 inactivation may not be complete and therefore could lead to an under-estimation of expression. Although (to my knowledge) it has not been investigated with neuronal/brain drivers, 24h at 29°C does not result in full induction with muscle drivers (microPublication Biology 2023. 10.17912/micropub.biology.000770.).

We have added a brief discussion of this point in the text (lines 277-280). We also note (lines 126-129) that the original TARGET publication (PMID 14657498) investigated the dynamics of induction of a GFP reporter in the brain using *elav[c155]-GAL4*, and found equivalent mRNA levels of the reporter after only 6 hours of induction, implying that 24 hours should be more than adequate. We note that the induction time of 24 hours was kept consistent throughout, meaning that the levels of reporter gene expression should reflect changes in the levels of GAL4, regardless of the exact dynamics here.

The first section of the results would benefit of some clarifications. ONLY the C155 driver is known to be described as pan-neuronal while ChAT and nSyb are not as far as I know. It is unclear what drivers the authors refer to as pan-neuronal when they write: "The GFP expression pattern of both pan-neuronal drivers was quite different from what we expected".

nSyb is also commonly used as a pan-neuronal driver in the literature (some recent high-profile examples: PMID 35121731, PMID 35869175, PMID 28604683). We have updated the text make this more explicit.

The reference de Chaumont et al., 2012 is not in the reference list.

We have updated that reference.

Only figure 1 shows the magnification scale, it is missing in Figures 2, 3, 4, 6 and all supplemental figures.

Magnification scale is the same across all images in the manuscript. We have added scale bars to all images for clarification.

The Data Availability Statement is missing (should be at the end of Material & Methods).

We have updated the text accordingly.

Reviewer #3

1. In the second last paragraph of the introduction, the authors state that *Drosophila* is becoming a powerful model organism ... I think it is better to rephrase this since *Drosophila* has been used for these types of studies since at least two decades.

We agree, and have edited the text accordingly.

2. In the last paragraph of the introduction, the authors write that the expression patterns in young adult brains were surprisingly not as uniform as expected. This statement is based on the assumption that it should be uniform, but is the evidence for that? *Elav* starts to be expressed in newly differentiating postmitotic neurons, and it may well have an initially higher expression that then goes down to a maintenance level. In other words, it may well be that this is the correct reflection of expression dynamics (that may not have been appreciated until now). At any rate, I suggest to carefully check the literature and possibly modify this statement.

A majority of papers involving *elav[c155]-GAL4* describe it as a pan-neuronal driver, with the underlying assumption that most neuronal cell types will express it in similar levels. Even though it has been the most common driver used in the field of fly neurobiology, its strong bias for expression in regions like the mushroom body has barely been mentioned. This lack of careful characterization of *elav[c155]-GAL4*, especially in the adult brain, together with its description as a “pan-neuronal” driver has likely misled many researchers into assuming it is expressed uniformly across the brain.

3. The authors use a shift to 29°C to induce expression of UAS reporters and this for 24 hours in all conditions. It is not clear to me whether the authors have checked the induction dynamics. Is 24h sufficient to get full expression (we typically use 48-72hrs of induction)? Is it certain that the induction dynamics itself does not change with age (certainly in light of overall reduced transcription at higher age)?

We routinely use 24 hours as induction time when performing Targeted DamID in the brain (e.g. PMID 29273756) and have only needed to use longer times (48 hours) when a small population of cells was targeted. The original TARGET publication (PMID 14657498), using *elav[c155]-GAL4*, observed equivalent mRNA levels after only 6 hours of induction. When we tested inductions of 48 hours, we observed slightly stronger expression levels, likely representing accumulation of the membrane-bound GFP reporter protein, but no changes in expression patterns.

To the best of our knowledge, we are not aware of any study looking at GAL4 induction dynamics in ageing. It is indeed possible that a general decrease in transcriptional rates during ageing might

affect transgene levels, and we have edited the text to mention this possibility (lines 293-296). We have also added additional discussion text on the possibility that longer induction times might lead to different expression profiles, while noting the consistency of our induction window throughout the study (lines 277-280).

Regardless of the exact mechanism, however, this does not change our conclusions regarding the change in GAL4 driver activity during ageing.

4. Please include in the materials and methods how long the brains were fixed. Since the authors do not perform antibody staining to visualize GFP, this information is important. Prolonged fixation tends to reduce fluorescence emitted by GFP.

We have edited the text accordingly, and we note that we used a consistent fixation procedure for all samples (including with respect to timings).

5. I do not understand why the fact that *elav-GAL4* is located on the X chromosome is an argument to only dissect male brains. One can equally well generate females.

Indeed, either males or females would be appropriate here; the main point here was that only one sex was used, as the expression pattern of *elav[c155]-GAL4* has previously been shown to be sex-dependent (PMID 26208119). Here we chose the sex with the stronger expression levels.

6. Did the authors use single optical sections in their analyses? Or did they use Z-stacks? Please clarify. If single optical sections were used, how was position in the brain controlled for?

We used maximum projections of Z-stacks. We have edited the text accordingly.

7. In the first part of the results, the authors write: 'Flies were reared at 18°C until 1DPE when brains were placed at 29°C ...' Were the brains first dissected and then placed at elevated temperature?

This was indeed misleading. The text has been edited to indicate that flies were reared at 18°C and then switched to 29°C for induction. After 24 hours at 29°C, brains were dissected.

8. It is remarkable that the authors were able to look at flies at age 90DPE. Usually, maximal life expectancy is about 60 days. Please explain.

We reared the flies at 18°C until the 24h induction, meaning that for the 90 DPE time point, flies were kept at 18°C for 90 days after eclosion. As lifespan is doubled at 18°C compared to 25°C (PMID 32134916), this time point is roughly equivalent to 45 days after eclosion at 25°C. We have edited the text to make these temperature dynamics and their effects on lifespan clearer to avoid confusion, and we have also edited the figures to explicitly show the temperatures used in our experimental setup.

9. Glial driver activity. The authors refer to the study by Sheng et al. Which glial GAL4 drivers did Sheng et al study? Did it include repo-GAL4? If not, how can repo-GAL4 be a good comparison?

The Sheng et al study used different glial drivers. However, repo-GAL4 is the most widely used driver for glia, so we felt it was important to characterise it as it would have a stronger impact in the field. We were unable to find any papers describing the expression dynamics of repo-GAL4. The Sheng et al study was the closest one we could find in relation to glia.

10. The authors use the Wissing et al (2022) reference, but as I understand it, this is a mouse reference. Are there no equivalent references for *Drosophila*?

To the best of our knowledge, there is no equivalent reference for *Drosophila*.

11. In the discussion, the authors refer to another pan-neuronal driver (also based on nSyb). Please include the reference!

This was a mistake on our side as we did not clearly write that the references provided in the first half of the sentence covers both nSyb drivers. We have edited the text to clarify this.

12. It would help the manuscript, if the authors would analyze the single cell expression data that were recently published for expression of elav, nSyb, ChAT, repo, and alphaTub84B and determine how this changes with age.

We have indeed checked the available scRNA-seq ageing datasets, but were not able to find conclusive results as the quality of the data decreases once we looked at separate ages individually. When grouping “young” vs “old” samples, we did not observe any clear difference. As discussed in the manuscript text, scRNA-seq data provides a relative, not absolute, measure of transcript abundance, and absolute changes in transcript levels may be missed using this technique.

13. In the final part of the discussion, the authors comment on all GAL4 drivers being very weak after only 30DPE at 18°C. Is this really so important? The majority of studies using *Drosophila* as model for neurodegeneration do not use flies older than 30 days, but more typically up to 21 days. Up until 28 days of age, flies tend to be fine, but then a fairly rapid decline in viability starts, making experimentation much more troublesome.

Flies aged at 18°C for 30 DPE are equivalent to flies aged at 15 DPE at 25°C, which is still an important time point for neurodegeneration and normal ageing studies. For example, studies on fly models for Alzheimer’s Disease typically measure effects on brain morphology or behaviour within a time window of 15 to 40 DPE at 25°C (i.e. 30 - 80 DPE equivalent at 18°C), and we note that this is only the most rapid-acting of neurodegenerative models. Parkinson’s (e.g. PMID 10746727) and ALS (e.g. PMID 33129345) models, for example, live significantly longer than AD models.

December 17, 2024

RE: GENETICS-2024-307300R1-A

Dr. Owen J. Marshall
University of Tasmania
Menzies Institute for Medical Research
17 Liverpool St
Hobart, N/A 7000
Australia

Dear Dr. Marshall:

Congratulations, your Brief Investigation entitled "Dynamic changes in neuronal and glial GAL4 driver expression during *Drosophila* ageing" is accepted for publication in GENETICS! Many thanks for submitting your research to the journal.

To Proceed to Publication:

1. Format your article according to GENETICS style: <https://academic.oup.com/genetics/pages/general-instructions>
2. Ensure that you comply with data and community resource citation guidelines:
<https://academic.oup.com/genetics/pages/general-instructions#Data-Policy>
3. Upload your final files at <https://genetics.msubmit.net>
4. Add oupsupport@scipris.com and genetics.oup@novatechset.com (or the domains @scipris.com and @novatechset.com) to your email program's "safe senders" list. You will be contacted by both at various points during the production process.

Notes:

- Your currently-accepted manuscript (unedited, as submitted, reviewed, and accepted) will be published at GENETICS and deposited into PubMed as an Advance Access article. Notify sourcefiles@thegsajournals.org before signing your license if you do not wish to publish your article via Advance Access.
- We invite you to submit an original color figure related to your paper for consideration as cover art. Please email your submission to the editorial office or upload it with your final files. You can submit a small-sized image for evaluation, and if selected, the final image must be a TIFF file 2513px wide by 3263px high (8.375 by 10.875 inches; resolution of 600ppi). Please avoid graphs and small type.
- After files are sent to Oxford University Press we use SciPris to manage article licensing and payment. If you do not have a SciPris account, you will receive an email from no-reply@scipris.com to sign up to use Oxford University Press' author portal. After logging in, follow the online instructions to sign your license and arrange any payment due.

If you have any questions or encounter any problems while uploading your accepted manuscript files, please email the editorial office at sourcefiles@thegsajournals.org.

Sincerely,

Karla Kaun
Associate Editor
GENETICS

Approved by:
Kate O'Connor-Giles
Senior Editor
GENETICS